# Costing interventions in the field: preliminary cost estimates and lessons learned from an evaluation of community-wide mass drug administration for elimination of soil-transmitted helminths in the DeWorm3 trial

Katya Galactionova [1,2] Maitreyi Sahu,[1,2] Samuel Paul Gideon,[3] Saravanakumar Puthupalayam Kaliappan,[3] Chloe Morozoff,[4] Sitara Swarna Rao Ajjampur,[3] Judd Walson,[4] Arianna Rubin Means,[4] Fabrizio Tediosi[1,2]

For numbered affiliations see end of article.

**Correspondence to**
Dr Katya Galactionova;
e.galactionova@unibas.ch

## ABSTRACT

**Objective** To present a costing study integrated within the DeWorm3 multi-country field trial of community-wide mass drug administration (cMDA) for elimination of soil-transmitted helminths.

**Design** Tailored data collection instruments covering resource use, expenditure and operational details were developed for each site. These were populated alongside field activities by on-site staff. Data quality control and validation processes were established. Programmed routines were used to clean, standardise and analyse data to derive costs of cMDA and supportive activities.

**Setting** Field site and collaborating research institutions.

**Primary and secondary outcome measures** A strategy for costing interventions in parallel with field activities was discussed. Interim estimates of cMDA costs obtained with the strategy were presented for one of the trial sites.

**Results** The study demonstrated that it was both feasible and advantageous to collect data alongside field activities. Practical decisions on implementing the strategy and the trade-offs involved varied by site; trialists and local partners were key to tailoring data collection to the technical and operational realities in the field. The strategy capitalised on the established processes for routine financial reporting at sites, benefitted from high recall and gathered operational insight that facilitated interpretation of the estimates derived. The methodology produced granular costs that aligned with the literature and allowed exploration of relevant scenarios. In the first year of the trial, net of drugs, the incremental financial cost of extending deworming of school-aged children to the whole community in India site averaged US$1.14 (USD, 2018) per person per round. A hypothesised at-scale routine implementation scenario yielded a much lower estimate of US$0.11 per person treated per round.

**Conclusions** We showed that costing interventions alongside field activities offers unique opportunities for collecting rich data to inform policy toward optimising

### Strengths and limitations of this study

► Resource use data were collected by on-site trial teams alongside field activities.
► This ensured high recall, supported collaboration between trial teams and grounded costing in operational insight.
► Separating data collection from the analysis, facilitated tailoring of tools without undermining consistency of the estimates between sites.
► While comprehensive in the scope of resources evaluated, the methodology was labour intensive to implement and required expertise to derive costs from the data collated.

health interventions and for facilitating transfer of economic evidence from the field to the programme.

**Trial registration number** NCT03014167; Pre-results.

## INTRODUCTION

Field trials conducted outside of clinical settings generate important evidence for making public health decisions and optimising health programmes. They indicate what the impact of efficacious interventions might be under real-world conditions, within representative target populations and with implementation processes that aim to inform routine programmatic activities.[1] Increasingly, economic outcomes, including costs and cost-effectiveness, are evaluated within community-based research studies to generate evidence regarding sustainability and scalability of tested interventions.[2–6]

While methodological guidance on economic evaluation and valuation of costs has been clearly outlined,[7–10] guidance on practical aspects of cost data collection in the field is relatively scarce.[11] Normative documents, including the recently released Reference Case for Global Health Costing,[10] cover some aspects of cost data collection including the perspective to take, the level at which to conduct costing and offer guidance on cost classification categories to adopt for presenting results of the study. However, apart from a brief overview of approaches for planning a costing exercise and potential data sources, these guidelines are limited in their description of strategies for integrating costing studies within field trials. A number of technical documents, targeted primarily at programme managers, offer guidance on budgeting of health interventions[9 12–15]; these programme-specific guidelines emphasise retrospective evaluation of interventions directed by costing templates or interactive tools. Guidelines for cost data collection along clinical trials, on the other hand, support ongoing costing activities.[16 17] Although primarily aimed at developed country contexts, these documents discuss a number of considerations relevant for field evaluations such as the need to isolate trial-driven resource use, the importance of prioritising data collection toward resource line items that are most likely to change the policy decision (ie, if costing is to support cost-effectiveness analyses) and integration of costing within the routine trial reporting. Finally, guidelines on costing health interventions in low-income and middle-income countries address specific challenges that arise due to poor data availability and limited economic expertise in these countries.[18 19] The guidelines further highlight the trade-offs between accuracy and data availability and suggest use of mixed methodologies (ie, gross costing along with bottom-up approaches) to tackle these limitations.

Integrating costing within a multi-country field trial raises basic practical questions not adequately addressed by this literature. How to assign roles for cost data collection and analysis, when to collect the data and for how long, what level of detail is appropriate and can be feasibly accommodated in the context of a field study, which tools to use, how to balance specificity of sites with the need for consistent comparison of costs between sites, how best to accommodate multiple economic endpoints and so on. Choices made by researchers to address these questions have important implications for the cost estimates derived and for their relevance for policy decisions the trial ultimately aims to inform. Somewhat surprisingly, there has been remarkably little exchange between practitioners on their experience with different strategies for costing interventions in the field.[19 20]

In this paper, we present our strategy for incorporating a costing study within the DeWorm3 project (ClinicalTrials.gov identifier NCT03014167)—a community-based cluster-randomised trial testing the feasibility of interrupting transmission of soil-transmitted helminth (STH) infections in India, Benin and Malawi.[2 21] This project is funded by the Bill & Melinda Gates Foundation through a grant awarded to the Natural History Museum (NHM). The DeWorm3 interventions consist of bi-annual community-wide mass drug administration (cMDA) targeting eligible individuals of all ages compared with standard of care deworming of school-age children (SAC). The trial is conducted in focal geographic areas with a population between 100 000 and 140 000 residing in each country site. cMDA is deployed for 3 years (six rounds) followed by 2 years of monitoring for recrudescence. The project aims to generate evidence for local and global stakeholders on optimal strategies for control and elimination of STH, including multi-level factors influencing treatment coverage.[21]

Although the trial and its evaluation, including costing, are ongoing, we share our early experience with the methodology to stimulate exchange among practitioners on the design and implementation of costing studies in the field. We discuss here some of the practical choices made and ways in which these decisions were motivated by the aims of the project, the infrastructure and the implementation of the trial interventions in specific sites. We report preliminary cost estimates for the India trial site and reflect on the challenges encountered in implementing our costing strategy, the ways in which we addressed these and highlight the lessons learnt for future costing studies operating in similar contexts.

## METHODS
### Interventions
The current global strategy for STH control relies on MDA in SAC.[22] The frequency and targeting of MDA campaigns are determined by the prevalence of infection in SAC with more frequent treatment where prevalence is high.[23] The Indian STH programme distributes MDA with Albendazole to pre-SAC and SAC children via a bi-annual national mass campaign called the National Deworming Day (NDD).[24] Drugs are administered during school days by teachers under the supervision of the local medical staff and the Ministry of Health (MoH). cMDA programmes for STH control, on the other hand, target eligible community members of all ages. In India cMDA in DeWorm3 intervention clusters started a week after NDD; drugs were administered to all individuals who were not treated in schools (ie, adults and out-of-school children). Prior to drug distribution, the DeWorm3 staff sensitised the community by distributing information, education and communication (IEC) materials, designed based on the NDD campaign with additional artwork depicting the treatment of whole communities, and by making public announcements. Community drug distributors (CDDs) were recruited from community volunteers and Accredited Social Health Activist workers. CDDs and project field officers supporting drug distribution were trained by the DeWorm3 team on consent seeking, drug administration and data collection. A medical team was trained on adverse event monitoring. During the

campaign rounds, CDDs were accompanied by project field officers and supervised by project staff. CDDs were paid a daily per-diem and received a mobile data package for each campaign round. Additionally, health facilities received an annual honorarium payment for health centre staff that supported the project on site. MDA coverage was assessed in both intervention and control clusters by field officers using mobile devices equipped with SurveyCTO.[25] Data were uploaded to a central database and analysed by the trial data management team. Mop-up drug distributions were conducted in all clusters in all rounds. A detailed description of operational activities and resource use supporting cMDA is presented in online supplemental tables S1 and 2.

### Study purpose, perspective and scope
The purpose of this study was to present our strategy for integrating costing into the DeWorm3 field trial.[21] To inform the broader aims of the project with respect to the cost-effectiveness of cMDA, the societal perspective was taken in the costing evaluation, and an assessment of both financial (most relevant for Neglected Tropical Disease (NTD) programmes and short-term planning) and economic costs (most relevant for MoH and global stakeholders) was undertaken (see online supplemental box S1 for key terms). Costs were evaluated around an operational intervention model following the micro-costing approach.[26] This methodology, while effort-intensive, grounded the evaluation in the implementation process and yielded granular data to inform modelling of cMDA modalities and contexts beyond those observed in the trial.

### Integration of costing within the DeWorm3 project
Integrating a costing study within a clinical trial required deciding who will collect the data, when and for how long. The DeWorm3 project included local institutions with strong epidemiological and clinical field research expertise (table 1). Capitalising on the interest of these organisations to develop capacity for economic evaluation, in-country teams were designated to lead cost data collection using standardised tools and with support from off-site health economics research groups. Ensuring country ownership was central to DeWorm3 design, as the project aims to facilitate transfer of knowledge from the field to support STH programmes and policies in countries hosting the trial.

DeWorm3 is a hybrid trial, which incorporates implementation and operational questions into a clinical trial design.[27] Since costs are an implementation outcome, economic analyses were considered under the umbrella of implementation science (IS) research (figure 1). The

| Location | Grant-holding organisation | Implementing organisation and partners | National standard of care STH control strategy (age target) | Community-wide STH MDA | Primary costing point person(s) |
|---|---|---|---|---|---|
| Come Commune, Benin | Institut de Recherche Clinique pour Développement | ▶ Institut de Recherche Clinique du Benin. ▶ Institut de Recherche Clinique pour Développement. ▶ Ministry of Health, Benin. | School-based MDA (5–14 years old) | Bi-annual community-wide MDA in all ages | Accountant, assistant accountant, IS lead |
| Tamil Nadu State, India | Christian Medical College, Vellore | ▶ Christian Medical College, Vellore. ▶ Ministry of Health and Family Welfare, India. | School-based MDA and National Deworming days (1–19 years old) | Bi-annual community-wide MDA in all ages; as a mop-up following NDD | Grant manager, trial coordinator |
| Mangochi District, Malawi | Blantyre Institute for Community Outreach | ▶ Blantyre Institute for Community Outreach. ▶ London School of Tropical Medicine and Hygiene. ▶ Ministry of Health and Education, Malawi. | School-based MDA and National Deworming days (1–14 years old) | Bi-annual community-wide MDA in all ages | Accountant, field accountant, field manager |

IS, implementation science; MDA, mass drug administration; NDD, National Deworming Day; STH, soil-transmitted helminth.

**Figure 1** Integration of costing study within the DeWorm3 trial research activities. The figure illustrates key surveys (top blue panel) and implementation science (IS) studies (bottom green panel) conducted along community-wide MDA in the DeWorm3 trial. It maps cost-effectiveness (to which costing contributes) within the IS research and highlights the contribution of other studies to costing and broader economic research agenda of the project. For further details refer to online supplemental table S1. Figure adapted from presentation by Means et al.[33] MDA, mass drug administration; IEC, information, education and communication materials; STH, soil-transmitted helminths.

IS studies collected data on operational processes, health systems and policy contexts, and community experience with cMDA. Thus, the data collection for the costing study could narrowly focus on direct financial and economic costs of cMDA to the project.

The costing study covered all field activities (figure 1, black and blue panels). This removed the need for cost data collection point persons at each site to decide which expenditures to record and allowed for a comprehensive assessment of resources that contributed to cMDA coverage and, ultimately, intervention effectiveness. Moreover, by tracking resources beyond operational activities immediately supporting cMDA, the study collated inputs for modelling costs of programmatic components such as surveillance and monitoring that are part of routine NTD control (ie, population census and prevalence surveys).

Cost data were collected over 2 years (four MDA rounds) in order to capture improvements in efficiency of the programme as teams gained operational experience deploying the intervention over multiple rounds.

### Integration of costing within the trial sites

The key feature of our study design is the delineation of responsibilities for data collection and valuation between the collaborators. Costing teams at each site were responsible for identifying resources used to support implementation of trial activities, extracting information on their respective prices and quantities, and recording these data using standardised data collection instruments. The off-site economics team, on the other hand, was responsible for triangulating these data to value resources in economic terms.

We conducted several rounds of training and formative meetings with site-designated costing point persons and other supporting staff. Several instruments, described briefly in table 2, were developed to guide cost data collection. These tools included an Excel-based Costing Tool, an Activity Table and an Activity Calendar. The Costing Tool detailed by activity quantities, prices and costs of the resources used, while the Activity Table described implementation of activities costed in the field covering additional resources for which no expenditures were incurred. Data entry templates from the Costing Tool and an example of a filled-out Activity Table for India site are shared in (online supplemental file S1 and table S2).

In the Costing Tool, resources were tracked differently by stage of implementation, specifically distinguishing between planning and other pretrial activities (ie, start-up) and activities taking place during the trial itself. For the start-up phase, sites reported the number of meetings, time dedicated to planning, sensitisation of country stakeholders, and development of training and sensitisation materials for the trial. In the implementation stage, in addition to resource lists and quantities, information was collected also on prices, expenditure and other details relevant to economic valuation (eg, year of purchase for capital items). The implementation modules of the Costing Tool further distinguished between resources that contributed to multiple trial activities (shared resources)

**Table 2** DeWorm3 instruments for collecting inputs to economic analysis

| Costing instrument | Type of cost | Content | Responsible person | Location of responsible person |
|---|---|---|---|---|
| Activity Table* | Financial and opportunity | Description of operational activities and subactivities, number of project staff and other resources used, number of days | Grant manager, trial coordinator | Field |
| Activity Calendar | | Start, end dates and duration of operational activities | Field manager, trial coordinator | Field |
| Costing Tool | Financial | Resource line items, corresponding prices, quantities and expenditure recorded by subactivity; separate modules for start-up and implementation | Grant manager, trial coordinator | Site |
| Spend Report | Financial | Trial expenditures | Project financial manager | External |
| Central Expenditure Report | Financial | List of prices and quantities of items procured at the central level | Project financial manager | External |

*An example of an Activity Table detailing a subset of trial activities implemented related to community-wide mass drug administration is enclosed in online supplemental table S2.

such as trial principal investigator (PI), office building, vehicles and so on from resources that contributed to a single activity. Shared resources were recorded in the Costing Tool at the start of the trial and were reviewed for completeness at each round of data collection. Thus, for each trial activity, only the incremental expenditures (ie, those additional to programme overheads and capital) were recorded by sites, thus minimising the effort and facilitating allocation of expenditure data.

Institutional differences and differences in technical capacity between sites required tailoring of the data collection strategy to each partner. Where reporting was centralised, one costing point person was designated with cost data collection, where different staff were responsible for dispensing funds for specific operational activities or resources (ie, fuel or bulk purchases of supplies) multiple staff assumed responsibility for compiling cost data at their respective levels. During piloting visits, the economics team, working together with in-country partners, mapped the costing strategy into sites' routine financial reporting, tailored to the site the costing instruments developed and updated standard operating procedures that outlined roles, responsibilities and reporting timelines for costing at the site.

As an illustrative example, figure 2 shows the flow of funds and financial reporting at DeWorm3 India site. The site designated their grants manager to lead cost data collection. Guided by the Costing Tool, the grants manager extracted information from receipts originating from the project's central offices, the Christian Medical College, Vellore which hosted the trial and the field sites. Expenditures that were incurred outside of the trial site (ie, bulk purchases of commodities and supplies by NHM DeWorm3 central team) were sourced and added to the Costing Tool in the analysis stage. Integrated within the trial operations, the costing study benefitted from site's

established financial reporting processes, quality control and oversight by partner institutions.

### From costing inputs to an estimate of intervention costs

Separating cost data collection from valuation of resources meant that the bulk of data cleaning, categorisation, consistency checks and validation happened during data quality and assurance checks at the central level. Entries in the Costing Tool were first evaluated for completeness and then for consistency by comparing expenditures and resource use against activities implemented as detailed in the Activity Table. During data quality checks, we standardised descriptions of resource line items, cost categories and activities to allow comparisons within and between different activities at each site. We checked for duplicate entries, ensured that the same resources were priced at the same rate, and those similar activities yielded comparable costs. Where inconsistencies were identified, site teams provided additional operational details and clarifications to resolve the issue, and brainstormed on whether adjustments to the analysis were needed to reflect the implementation.

Once cleaned, we applied standardised programming routines, implemented in Stata 16.1,[28] to convert these data into estimates of cMDA and trial costs. Specifically:

1. *Start-up costs*: we retrospectively estimated costs of planning and other pretrial activities for which no expenditure data were recorded. In the Costing Tool these were described according to the type (ie, sensitisation meetings, micro-planning) and intensity of activities conducted (ie, the number of meetings). We used information from other sections of the Costing Tool (ie, project staff wages, transportation) to value the respective resource use via direct allocation. Start-up costs were treated as capital items and were annualised over the 5-year duration of the trial.

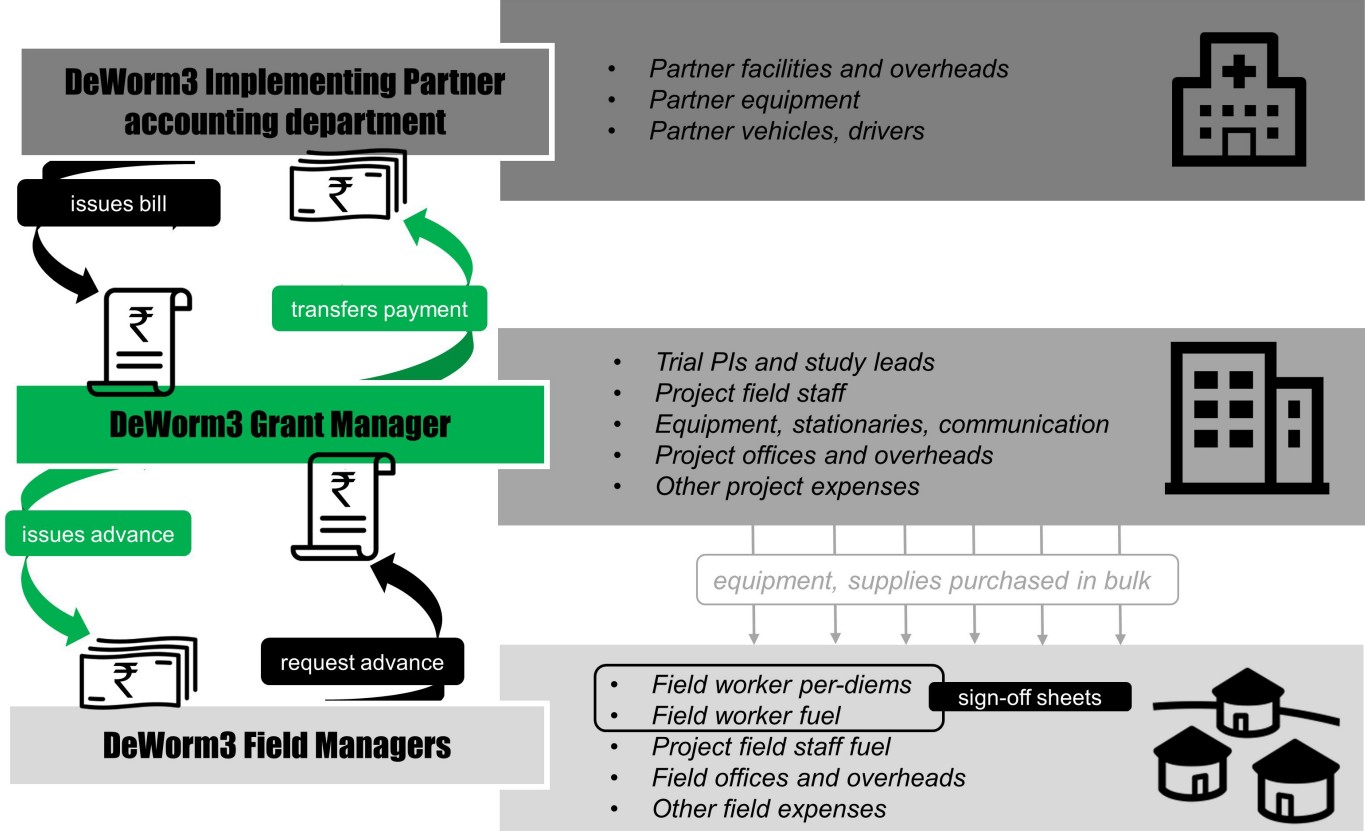

**Figure 2** Routine financial reporting at India DeWorm3 trial site. The figure illustrates the flow of funds and financial reporting at India DeWorm3 trial site. Project's grants manager—centre green box—is the focal point through which all country funds supporting the trial flow. For each project level the diagram shows staff responsible for dispersing funds (left) and gives examples of some of the resource line items that are purchased/consumed/recorded at that level (right panels). Bottom row—field site—additionally highlights itemised tracking of expenditures related to field worker per-diems and fuel (important cost driver of mass drug administration activities). PI, principal investigator.

2. *Activity costs*: costs of trial operational activities were obtained by summing costs of the resource line items recorded in the Costing Tool and adding on shared resources and programme overheads. Resource line items were valued according to their respective cost category (online supplemental table S3) applying, where appropriate, annualisation and adjusting for percentage used for the trial to estimate the value of these resources for the first cMDA cycle.

3. *Shared costs*: shared resources that could be allocated to specific activities and subactivities such as laboratory equipment were allocated accordingly, while the rest (primarily research and management staff and overheads) were grouped under 'Programme management' cost category (online supplemental table S4). Resources that contributed to multiple activities, such as some transportation-related expenses were allocated based on the number of activity days.

To derive the cost of cMDA, we isolated cMDA-related expenditures from all other trial expenditures. For the most part, the distinction between intervention and research costs was tied to activity or subactivity category (ie, expenditures related to training of CDDs supporting MDA were categorised as cMDA-related). If, however, an activity or a resource category contributed to both cMDA and research activities, in some sites, was the case for training of field workers, costs were allocated based on the ratio of cMDA days to total trial days. For some shared resources, costs were allocated multiple times: first to the respective activity and then to cMDA or the trial (online supplemental table S4). Total cost of cMDA was then obtained by summing over subactivities supporting drug administration, and adding on cMDA-attributable share of planning and programme costs.

Differences in implementation of the costing approach between sites required further tailoring of these standardised valuation and allocation routines. In India, for instance, compensation of field staff and some other field expenditures (ie, materials and supplies, some fuel and transportation costs) were tracked on a monthly, rather than weekly or daily, basis. These monthly expenditure totals often covered multiple activities each requiring a varying number of field staff. To allocate these, we first calculated a daily equivalent value for each expenditure type and where possible, staff category. Specifically, monthly expenditures were summed over the 12-month cycle within which the two rounds of cMDA were delivered and divided by the total number of staff, compensated

each month times 20.5 (the average number of working days per month according to India team). We then mapped these monthly expenditure categories into trial activities based on Activity Table, flagging activities to which a specific staff or resource category contributed. Finally, activity totals were obtained by multiplying the daily equivalent value for each expenditure type by the number of staff and the number of days over which the respective activity was conducted. To the extent that multiple activities, including those research-related, were conducted simultaneously by the field staff this strategy might have overstated total cMDA-attributable spending.

### Costs of cMDA in the DeWorm3 India site
To support a critical assessment of the study design presented, we share preliminary costs for the first year of cMDA in the DeWorm3 India site. These are interim findings that only integrate financial expenditures, final cost estimates from DeWorm3 are forthcoming. The incremental financial costs of cMDA were summarised

by key operational activities, including start-up and programme overheads. We additionally considered how our results from the trial might translate to costs under routine programmatic implementation by Indian NDD programme.

### Patient and public involvement
No patient involved.

### RESULTS
Table 3 presents interim results for two rounds of cMDA and supportive activities conducted at the India DeWorm3 site between October 2017 and September 2018. In the first year, the cost averaged about US$1.14 per person treated per round. While the unit costs were virtually identical between the two rounds, there were differences in the number of field staff (lower in the second round) and the number of implementation days for sensitisation, training and drug distribution campaign (higher in the

**Table 3** Interim results from first year of STH cMDA* implementation in India DeWorm3 site: financial costs, USD 2018

| | Total (US$) | | | Average cost profile (%) | Average cost per person treated per round (US$) | Average cost per capita per round (US$) |
|---|---|---|---|---|---|---|
| MDA round | 1 | 2 | 1+2 | 1+2 | 1+2 | 1+2 |
| Total | 60 320 | 54 305 | 114 626 | 100.0 | 1.14 | 0.84 |
| Total incremental† | 4829 | 6577 | 11 406 | *10.0* | *0.11* | *0.08* |
| Start-up activities‡ | 1094 | 1055 | 2149 | 1.9 | 0.02 | 0.02 |
| Programme management | 21 300 | 20 540 | 41 840 | 36.5 | 0.42 | 0.31 |
| Drug testing and distribution to site | 304 | 331 | 636 | 0.6 | 0.01 | 0.00 |
| CDD recruitment | 1066 | 1028 | 2095 | 1.8 | 0.02 | 0.02 |
| Community sensitisation | 12 924 | 9343 | 22 267 | 19.4 | 0.22 | 0.16 |
| CDD bags and job aids | 555 | 428 | 983 | 0.9 | 0.01 | 0.01 |
| Banners, posters | 969 | 969 | 1937 | 1.7 | 0.02 | 0.01 |
| Training | 5003 | 2840 | 7843 | 6.8 | 0.08 | 0.06 |
| Travel allowance for CDDs | 240 | 170 | 409 | 0.4 | 0.00 | 0.00 |
| Refreshments for CDDs | 10 | 17 | 26 | 0.0 | 0.00 | 0.00 |
| cMDA | 13 442 | 14 659 | 28 101 | 24.5 | 0.28 | 0.21 |
| CDD incentives | 2632 | 4577 | 7210 | 6.3 | 0.07 | 0.05 |
| CDD mobile allowance | 120 | 85 | 205 | 0.2 | 0.00 | 0.00 |
| Coverage survey | 3116 | 2451 | 5567 | 4.9 | 0.06 | 0.04 |
| Mop-up | 2070 | 2058 | 4128 | 3.6 | 0.04 | 0.03 |

Grey shaded rows are a subset of the higher level activity grouping. Total number of people treated in first round of cMDA (cMDA1) was 51 320 (site total population 68 442); in second round (cMDA2)—49 488 (site total population 68 460); total treated over the two rounds (cMDA1 +cMDA2) was 100 808 (total population 136 902). Costs were converted to USD using average annual exchange rate over the study period (1 INR=US$0.01462).[34]
*cMDA was implemented as a mop-up following NDD campaign that targeted school-aged children.
†Incremental costs represent a subset of rows highlighted in italics, see text for details.
‡Start-up activities annualised over the duration of the trial (5 years).
CDD, community drug distributor; cMDA, community-wide mass drug administration; STH, soil-transmitted helminth infections.

second round). Drug administration in the community accounted for over a quarter of total intervention cost (row 'cMDA', table 3). As drugs were donated, cMDA costs were primarily driven by personnel: wages of field staff and incentives to CDDs accounted of about 70% of total costs (online supplemental table S5). The second-largest cost component was 'programme management and overheads' (33% of total cost). This cost category covers wages of core trial personnel and expenses related to transportation for supportive supervision or other activities conducted by this management cadre. Programme overheads also include data processing and data management costs (ie, related to registration of households for MDA).

Due to the large contribution of field staff to the overall cost of cMDA, the estimates were highly sensitive to decisions on how these resources were quantified and valued. The estimates presented in table 3 were obtained by applying a fixed monthly base rate to the number of staff and activity days to derive total cost. In practice, though, there was some variation around the amount paid to field staff. At the India site, field workers were borrowed from other studies to assist in drug distribution. These additional staff were compensated at their regular rate which was below the rate paid in the DeWorm3 project. Moreover, during the campaign, field staff also worked on weekends and through holidays earning overtime. To better capture the value of field staff we re-ran the analysis applying an adjustment weight to the base wage. The site staff-level weight was calculated as a ratio of the respective monthly base rate to monthly equivalent rate derived from site expenditure data. It varied from 0.9 to 2.5 between staff categories: nearly identical to the monthly base rate for field officers (weights ranging between 0.9 and 1.01) and substantially above the base rate for field managers and supervisors indicating overtime. One weight was then applied to all subactivities that started in a given month thus inflating base wages depending on the total expenditure incurred in a given month. After this rate adjustment the average cost per person treated per round increased to US$1.20 (online supplemental table S6).

Arguably, from the perspective of the programme the primary interest is in the incremental resource needs for reaching adults and children not treated under the current NDD deworming campaign. Thus, the programmatic relevance of the totals reported for the trial above and some of the resource line items costed warrant further consideration. For example, it could be suggested that planning, programme management and community sensitisation activities would not necessarily change to accommodate the broader age target for cMDA. Provided there is spare capacity in the system, costs related to drugs, drug storage and distribution, and also coverage surveys and mop-up, would scale linearly with the number of people covered. The incremental costs, compared with the current SAC-targeted strategy, would then be incurred for recruitment, training and incentives of/to CDD's (see table 3 rows highlighted in cursive). This scenario adds up

to a significantly lower cost estimate per person treated of US$0.11, compared with US$1.14 reported from the perspective of the trial. Further analyses with input from the programme are needed to appropriately contextualise these findings.

Recognising that comparisons of costs between studies are often limited due to differences in the objectives of respective evaluations, scope of resources costed as well as methodological choices made by analysts,[10] these comparisons, nonetheless, provide additional support to the methodology described here. We found that the cost estimates derived align reasonably well with cMDA costs reported elsewhere. For instance, a recent multi-site study from African region reported an average of US$1.75 per person treated per round of cMDA treatment.[29] A lower estimate of US$0.65 (Niger GDP deflator applied to 2005 estimate of US$0.46 reported in the study[10]) per person treated was reported over a 2-year programme in four districts in Niger, this is about half the estimate of drug delivery for SAC in the same study.[30] The scale of the programme: 300 000 to over 450 000 people treated in Niger compared with just under 70 000 in the DeWorm3 India site explains some of the difference in costs between the two studies. Another estimate comes from a meta-regression of MDA costs for NTDs; net of community volunteers and drugs, the estimate averaged US$0.40 (range US$0.02–US$2.90) per person per round,[31] which is comparable to US$0.58 in this study when we exclude payments to CDDs and field officers.

## DISCUSSION

The costing strategy presented here was developed in close collaboration with trialists both at and off-site that helped us adapt it to the technical and operational realities at sites. The methodology capitalised on the immediacy of resource use on the ground, the operational expertise and established processes for routine financial reporting within the implementing organisations. Integrating costing within the DeWorm3 trial allowed us to build on other research studies conducted concomitantly to get a fuller appreciation of resource use related to cMDA. Implementing this strategy required a multitude of practical decisions that were necessarily context specific and entailed trade-offs between precision and effort. We carefully consider these below to inform choices analysts make when designing costing exercises in similar settings. We hope to encourage exchange among practitioners on their experience with different methodologies for costing interventions in the field.

### Balancing flexibility and consistency in designing the strategy

One of the key challenges in conducting a multi-country cost analysis is in balancing the need for flexibility to accommodate the trial and intervention features in a given setting with the need for consistency and standardisation of cost estimates across the trial sites. Our reliance on the micro-costing methodology and use of multiple instruments

to detail operational activities on the ground, including resource inputs, helped capture these differences. Piloting of the tools, and their tailoring to and integration into routine financial reporting further ensured that robust estimates could be derived despite differences in sites' contextual features. Adapting data collection to sites, however, did complicate the analysis. We could ensure consistent valuation of resources by relying on programmed routines that fixed *how* resources were categorised, priced, annualised and allocated to the intervention across sites.

### Harnessing synergies and fostering collaborations within the project

The complexity of the DeWorm3 project posed a number of own challenges for the design and implementation of the costing study. With many research activities conducted alongside the intervention (eg, annual census, prevalence surveys), developing a costing strategy required coordination between the many project partners on and off-site. From decisions on who will lead the cost data collection to when to pilot the costing tools, to how best to monitor quality of the data collected—all had to be agreed on between multiple parties. The communication overhead was minimised by integrating the costing study under the broader IS research agenda; additionally, reporting was further streamlined by centralised tracking of costing outputs by the IS team which formally tied costing deliverables to funding transfers to sites.

While in the design stages of the project, we could identify synergies between studies, differences in the timing of data collection and processing meant that we could not fully avoid some duplication of effort. For instance, Activity Table compiled by field staff (ie, field manager or trial coordinator) at each site collated some of the same information as IS process mapping exercises. Activity Table filled out by the field staff during each trial activity was immediately available to the costing point person at the site and the economic teams at the submission of costing instruments; the IS outputs, on the other hand, populated with data from multiple study clusters, required time for data collection and processing before these could be integrated. At the same time, the overlap between data collection instruments allowed for triangulation of data in the analysis stage, which further strengthened the quality of the evidence.

The success of our costing strategy further relied on close cooperation between the different study leads at the site. Efficient communication and management within the implementing agency meant that it was possible for the designated field staff and costing point persons to validate resource allocation to specific activities and reflect those appropriately in the costing instruments. When sites were limited for time in collecting or reviewing costs, higher levels of aggregation of expenditure data were recorded.

### Challenges and opportunities in country-led cost data collection

We found that we could support site data collection efforts best when our guidance was grounded in an understanding of operational details of interventions being evaluated and the specific contexts in which the evaluation was conducted. To this end, delaying site visits and piloting of tools until field activities have commenced appeared to have been one of the critical choices we made in implementing the strategy. This ensured that both financing channels and routine reporting at sites were established by the time of the visit and we could map these together with costing leads to guide where and how best to collect the data.

Our initial guidance on selecting the focal person for cost data collection was an on-site team member that was primarily field-based and had a part-time duty to populate cost data collection instruments. However, due to the sensitivities of financial reporting, particularly of salaries, it was challenging to designate anyone other than accounting staff to collate economic data. This naturally imposed a substantial burden on accountants already fully engaged in their primary duties. To make it possible for an accountant, removed from the field, to lead cost data collection it was necessary to develop additional instruments (ie, Activity Table) and to facilitate a closer involvement of field staff with the costing study (ie, via Activity Table, Activity Calendar) than originally anticipated. On the other hand, these additional tools enabled unique insights into the implementation of interventions on the ground and facilitated quality control and subsequent analyses of the data collected.

Routine financial reporting at sites offered some unanticipated opportunities for cost data collection. For example, in Malawi, weekly financial advances were made by an off-site accountant to the field team which were reconciled the following week against a filled-out Expenditure Ledger. The latter, organised in an Excel table, itemised field expenses detailing prices, quantities along with a brief description for each entry. Both the frequency of recording and the level of detail at which expenditures were described in the Ledger made it easy to integrate this site-developed tool into the costing strategy.

Some of the project research studies were conducted either concomitantly with or relying on resources that also supported cMDA. Allocation of shared resources is an area that requires value judgments which are often difficult for those new to costing. There are several aspects of our strategy that made these choices less arbitrary. First, extending the scope of the evaluation to all trial activities (including research-related) removed the need for local teams to decide which activities or resources to track. Second, capital resource line items and other resources that contributed to multiple activities like programme staff were explicitly identified and tracked separately in the Costing Tool, thus focusing data collection on tracking incremental expenditures for each activity and leaving allocation of shared resources to the economics team.

Separating data collection and valuation of resources ensured that inconsistencies in data entry could be resolved in the analysis stage. This feature proved

advantageous as, despite multiple points of validation at the field site, entries in the Costing Tool were often mislocated between activities. These challenges highlight limitations of pre-populated costing tools that collate aggregated expenditures recorded by field teams into an estimate of unit costs. We were able to resolve these errors by triangulating information across the costing instruments and by comparing entries between the trial sites; being able to follow-up with teams shortly after an activity was completed to validate any corrections to the data ensured high recall.

Sites also struggled with the level of detail for cost data collection. Modifying internal reporting templates to include information on prices, quantities and activities to which an expenditure/resource line item contributed made it easier for sites to capture these data. However, it required more time to fill out these documents. Where there were multiple financially responsible persons the volume of receipts, particularly those attesting to petit-cash spending or other low-level expenditures, created a substantial hurdle in recording and transferring these data into costing tools.

Sites voiced difficulties mapping expenditures in the Costing Tool into their internal accounting documents. To allow sites to double-check expenditure totals and to ensure that expenditures were accurately and comprehensively reported in both systems we included an additional column in the Tool for sites' internal transfer number. These linked resource line items in the Tool to the internal financial tranche that paid for it, which further improved reporting.

We found that our approach worked best in sites where there was an established and lean process for financial reporting within the implementing organisation. Where bureaucracy was heavy, for instance, when reporting was required for multiple funders each with own accounting system there was low capacity to support data collection for the costing study. Multiple staff were required to facilitate the process and to transfer financial and resource use data into a format that could be used for the economic analysis.

### Relevance to programmes

The cost estimates of cMDA implementation presented here are only the first step in the economic analysis. Most immediately, the scope of the evaluation will expand to consider the full economic costs of the programme and broaden the perspective to include indirect costs to the community and the project by triangulating data from surveys conducted along the trial (figure 1). Informed with IS research, costing will further explore cost drivers and heterogeneity in cost of cMDA within and between sites. In India, the analysis will compare costs between two study subsites that differ in their population, sociocultural and economic characteristics, and geography. IS research will further help identify the incremental resource use for current and alternative cMDA implementation scenarios to feedback to programmes.

The operational model for cMDA implementation in the trial detailed in Activity Table forms the basis for an assessment of the delivery modality within the project to what delivery might look like when scaled up by the programme. Using the data collected, we can begin to isolate aspects of intervention design that are trial-driven and update these with assumptions reflecting operational realities of the programme. When presenting cost estimates for the trial we hypothesised the potential scope of incremental resources that might be relevant to India NTD programme, the resulting estimate, describing primarily incentives to CDDs and community sensitisation aids was drastically different than the one describing the full scope of resources that supported cMDA implementation in the trial (ie, US$0.11 compared with US$1.14 per person treated per round). The methodology thus facilitates careful interpretation of the costs derived and supports effective communication of findings to the national programme recognising the many ways in which the scalable model might differ from the one piloted and the likely cost implications of these differences.[17 32]

### CONCLUSION

Field research produces important insights about effectiveness of interventions in the community. The programmatic relevance of field trials can be further strengthened by integrating costing within the trial study design. We showed here that collecting cost data along field activities is both feasible and immediately informed by the implementation of interventions on the ground. The strategy presented offers unique opportunities for collecting granular data to support a broad range of policy questions toward optimising health interventions and facilitating transfer of economic evidence from the field to the programme.

**Author affiliations**
[1]Department of Epidemiology and Public Health, Swiss Tropical and Public Health Institute, Basel, Switzerland
[2]Faculty of Medicine, University of Basel, Basel, Switzerland
[3]Wellcome Trust Research Laboratory, Division of Gastrointestinal Sciences, Christian Medical College Vellore, Vellore, Tamil Nadu, India
[4]Department of Global Health, University of Washington, Seattle, Washington, USA

**Acknowledgements** The authors wish to thank all of the study participants, communities and community leaders, national NTD program staff and other local, regional and national partners who have participated or supported the DeWorm3 study. We also would like to acknowledge the work of all members of the DeWorm3 study teams and affiliated institutions. We give special thanks to Wongani Lungu, Chikondi Chikotichalera, Roselyn Hara, Euripide Avokpaho, Nadege Aicheou, Monrenike Bada, James Simwanza, David Chinyanya, Rajesh Kumar Rajendiran, Chinnaduraipandi Paulsamy, Elodie Yard, Stefan Witek-McManus, Katherine E Halliday, Hugo Legge, Leanne Doran, William E Oswald, Fabian Schär, Marie-Claire Gwayi-Chore, Mira Emmanuel-Fabua, Iain Gardiner, Moudachirou Ibikounle, Khumbo Kalua, Adrian J F Luty, and Robin Bailey.

**Contributors** KG, ARM, FT designed data collection tools. KG piloted data collection tools. SPG, SPK collected data and contributed to design of data collection tools. CM, MS conducted data queries. KG developed data analysis and validation routines. KG, CM analysed the data. KG wrote the first draft. KG, ARM, CM wrote the manuscript. MS, FT provided critical inputs and edits to the manuscript. KG, MS,

SPG, SPK, CM, SSRA, JW, ARM, FT reviewed and approved the final version of the manuscript for publication.

**Funding** This project was funded by a subcontract from the Natural History Museum grant from the Bill and Melinda Gates foundation OPP1129535.

**Competing interests** None declared.

**Patient consent for publication** Not required.

**Provenance and peer review** Not commissioned; externally peer reviewed.

**Data availability statement** Data (i.e. disaggregated costs and quantities by resource line item, Stata do-files) are available upon reasonable request. For inquiries contact Dr. Arianna R. Means at aerubin@uw.edu.

**ORCID iD**
Katya Galactionova http://orcid.org/0000-0002-5743-7647

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
