## [Reviewer comments · BMJ Open]

ARTICLE DETAILS

TITLE (PROVISIONAL)	Costing interventions in the field: preliminary cost estimates and lessons learned from an evaluation of community-wide Mass Drug Administration for elimination of Soil-Transmitted Helminths in the DeWorm3 trial
AUTHORS	Galactionova, Katya; Sahu, Maitreyi; Gideon, Samuel; PUTHUPALAYAM KALIAPPAN, SARAVANAKUMAR; Morozoff, Chloe; Ajjampur, Sitara; Walson, Judd; Rubin Means, Arianna; Tediosi, Fabrizio

VERSION 1 – REVIEW

REVIEWER	Fortaleza, Carlos Magno C. B. Faculdade de Medicina de Botucatu, UNESP - Univ Estadual Paulista, Tropical Diseases
REVIEW RETURNED	28-Feb-2021

GENERAL COMMENTS	The authors present a “side study” of a community trial originally focused on the efficacy and cost-effectiveness of applying massive anthelmintics in a community in India. Their objective here is to demonstrate the advantage of collecting cost data alongside field activities (rather than retrospectively or indirect estimation). The study was coherent, with a robust and detailed methodology. I have minor comments on the study below: 1. In the abstract the authors state that the study outcome measure was “Incremental cost of extending deworming of school-aged children to the whole community”. Well, that seems to be the outcome for the original study, or rather a secondary outcome in this paper, which is focused on the feasibility of collecting cost data.2. The summary correctly points to study relevant, findings and limitations.3. The introduction is clear and concise.4. In the methods section, the sub-section “Study purpose, perspective and scope” describes a broader objective that it really pertains this manuscript. While all those extensive analyses are part of the original project, some indication of the specific objective of this “side-study” should be emphasized.5. Still in the methods section, the techniques for operationalizing the “costing tool” are extensively described, allowing the reader to have a deep understanding of the rationale and procedures.6. Results and discussion are appropriate. I think that Table 3 should be emphasized in its preliminary and collateral aspect. It is true that its title reports “first year results”, but the relevance here is to demonstrate the authors point on cost collection tools, so that information should be available without necessary reference to the text.
--

REVIEWER	Manoukian, Sarkis Glasgow Caledonian University, School of Health and Life Sciences
REVIEW RETURNED	18-Mar-2021

GENERAL COMMENTS	This articles focuses on an important issue which is costing strategies in large trials. I found the article well-written however there are some areas of improvement. Abstract: I found this section confusing as part of it was about the trial rather than this paper. I would like to see a new abstract written specifically for this paper that clearly explains what this paper does. Summary: I do not think "high-resolution data" is correct terminology. I know what you are trying to say but this term needs revision. Costing-tool and STATA codes: I appreciate the fact that you have written this paper to help other researchers. In order to maximise the benefit and impact of your paper my suggestion is to make examples (if not all) available as supplementary materials of the Excel files and STATA codes you have used. This will allow researchers to be able to fully understand your strategies. Limitations: I would like to see a section about limitations rather than a collection of sentences here and there. Could you include the trial registration information somewhere if applicable? Discussion section: This section requires a bit of work, I found it a bit hard to read especially in the middle which seemed like reading a methods sections. I'd like to see this section revised. I consider this comment minor so I do not expect to see major revisions here just improvements to help the readers make sense of your results. I think you also need to explain more clearly what are the implications for other researchers before this is accepted for publication.
---

VERSION 1 – AUTHOR RESPONSE

Reviewer Reports:

Reviewer: 1

Dr. Carlos Magno C. B. Fortaleza, Faculdade de Medicina de Botucatu, UNESP
- Univ Estadual Paulista

Comments to the Author:

The authors present a “side study” of a community trial originally focused on the efficacy and cost-effectiveness of applying massive anthelmintics in a community in India. Their objective here is to demonstrate the advantage of collecting cost data alongside field activities (rather than retrospectively or indirect estimation). The study was coherent, with a robust and detailed methodology. I have minor comments on the study bellow:

1. In the abstract the authors state that the study outcome measure was “Incremental cost of extending deworming of school-aged children to the whole community”. Well, that seems to be the outcome for the original study, or rather a secondary outcome in this paper, which is focused on the feasibility of collecting cost data.

Author response:

We thank the Reviewer for this comment. We have now edited the abstract to appropriately reflect the focus of this paper on presenting our strategy for integrating a costing study within the DeWorm3 field trial.

2. The summary correctly points to study relevant, findings and limitations.

Author response:

We thank the Reviewer for this observation.

3. The introduction is clear and concise.

Author response:

We thank the Reviewer for this observation.

4. In the methods section, the sub-section “Study purpose, perspective and scope” describes a broader objective that it really pertains this manuscript. While all those extensive analyses are part of the original project, some indication of the specific objective of this “side-study” should be emphasized.

Author response:

We have now clarified the purpose of this study to present our strategy for costing cMDA and to distinguish the purpose of this study from the broader agenda of the DeWorm3 trial.

5. Still in the methods section, the techniques for operationalizing the “costing tool” are extensively described, allowing the reader to have a deep understanding of the rationale and procedures.

Author response:

We thank the Reviewer for this observation. Indeed, it is our hope that learnings from this study inform design and implementation of future costing exercises.

6. Results and discussion are appropriate. I think that Table 3 should be emphasized in its preliminary and collateral aspect. It is true that its title reports “first year results”, but the relevance here is to demonstrate the authors point on cost collection tools, so that information should be available without necessary reference to the text.

Author response:

We have edited the title of Table3 to emphasize the interim status of the costing evaluation at which our findings are reported.

Reviewer: 2

Dr. Sarkis Manoukian, Glasgow Caledonian University

Comments to the Author:

This articles focuses on an important issue which is costing strategies in large trials. I found the article well-written however there are some areas of improvement.

Abstract: I found this section confusing as part of it was about the trial rather than this paper. I would like to see a new abstract written specifically for this paper that clearly explains what this paper does.

Author response:

We thank the Reviewer for this comment. We have now edited the abstract to appropriately reflect the focus of this paper on presenting our strategy for integrating a costing study within the DeWorm3 field trial.

Summary: I do not think "high-resolution data" is correct terminology. I know what you are trying to say but this term needs revision.

Author response:

We have now removed all instances of "high-resolution data" in the manuscript and replaced the term with more precise wording emphasizing the level of detail and the comprehensive scope of the costing exercise.

Costing-tool and STATA codes: I appreciate the fact that you have written this paper to help other researchers. In order to maximise the benefit and impact of your paper my suggestion is to make examples (if not all) available as supplementary materials of the Excel files and STATA codes you have used. This will allow researchers to be able to fully understand your strategies.

Author response:

We thank the Reviewer for this comment. In the Supplementary Materials we included the templates used to collect resource use and other inputs in the trial: "File 1. DeWorm3 Costing Tool data collection templates". These are generic and can be readily used in another trial. Stata processing scripts, on the other hand, reflect the idiosyncronities related to data entry at particular sites (i.e. specification of units, rates, etc.) and their alignment to achieve consistency in the estimates derived. We believe the code itself is of limited use outside of the site. However, we have no reservations about sharing our code and added a statemet to the SI file that Stata code can be provided on request from the authors.

Limitations: I would like to see a section about limitations rather than a collection of sentences here and there.

Author response:

We thank the Reviewer for this comment. We have extensively deliberated on the best way to present the multitude of practical decisions involved in integrating a costing study within a field trial and the trade-offs one weighs when pursuing a particular strategy. These decisions are necessarily context specific – what might be a limitation in a fairly well-resourced setting is a clever strategy that yields reliable data in another. We believe that presenting limitations in the context of the features of the site and motivating why a particular strategy was pursued here is most helpful for informing decisions for future costing studies in similar contexts. This was now clarified in the opening paragraph of the discussion section.

Could you include the trial registration information somewhere if applicable?

Author response:

We have now added trial registration number to the description of the study.

Discussion section: This section requires a bit of work, I found it a bit hard to read especially in the middle which seemed like reading a methods sections. I'd like to see this section revised. I consider this comment minor so I do not expect to see major revisions here just improvements to help the readers make sense of your results. I think you also need to explain more clearly what are the implications for other researchers before this is accepted for publication.

Author response:

We thank the Reviewer for this comment, we have further edited the discussion section for clarity.

Reviewer: 1

Competing interests of Reviewer: None declared

Reviewer: 2

Competing interests of Reviewer: None declared

VERSION 2 – REVIEW

REVIEWER	Manoukian, Sarkis Glasgow Caledonian University, School of Health and Life Sciences
REVIEW RETURNED	15-Jun-2021
GENERAL COMMENTS	Accepted for publication